# High-throughput quantitation of inorganic nanoparticle biodistribution at the single-cell level using mass cytometry

Yu-Sang Sabrina Yang[1,2], Prabhani U. Atukorale[3,*], Kelly D. Moynihan[2,3,*], Ahmet Bekdemir[4], Kavya Rakhra[1,2,3], Li Tang[1,2,3], Francesco Stellacci[4] & Darrell J. Irvine[1,2,3,5,6]

Inorganic nanoparticles (NPs) are studied as drug carriers, radiosensitizers and imaging agents, and characterizing nanoparticle biodistribution is essential for evaluating their efficacy and safety. Tracking NPs at the single-cell level with current technologies is complicated by the lack of reliable methods to stably label particles over extended durations *in vivo*. Here we demonstrate that mass cytometry by time-of-flight provides a label-free approach for inorganic nanoparticle quantitation in cells. Furthermore, mass cytometry can enumerate AuNPs with a lower detection limit of ∼10 AuNPs (3 nm core size) in a single cell with tandem multiparameter cellular phenotyping. Using the cellular distribution insights, we selected an amphiphilic surface ligand-coated AuNP that targeted myeloid dendritic cells in lymph nodes as a peptide antigen carrier, substantially increasing the efficacy of a model vaccine in a B16-OVA melanoma mouse model. This technology provides a powerful new level of insight into nanoparticle fate *in vivo*.

[1] Department of Materials Science and Engineering, Massachusetts Institute of Technology, Cambridge, Massachusetts 02139, USA. [2] Koch Institute for Integrative Cancer Research, Massachusetts Institute of Technology, Cambridge, Massachusetts 02139, USA. [3] Department of Biological Engineering, Massachusetts Institute of Technology, Cambridge, Massachusetts 02139, USA. [4] École Polytechnique Fédérale de Lausanne, Institute of Materials and Interfaculty Bioengineering Institute, Lausanne 1015, Switzerland. [5] Howard Hughes Medical Institute, Chevy Chase, Maryland 20815, USA. [6] Ragon Institute of MGH, MIT and Harvard, Charlestown, Cambridge, Massachusetts 02129, USA. * These authors contributed equally to this work. Correspondence and requests for materials should be addressed to D.J.I. (email: djirvine@mit.edu).

norganic nanomaterials are employed clinically as imaging contrast agents and are under development for a broad range of additional biomedical applications[1]. Examples include iron oxide[2], platinum[3] and bismuth[4] based nanoparticles used as contrast agents in MRI and X-ray imaging; and gadolinium[5] and gold nanoparticles[6,7] being developed as radiosensitizers and drug delivery systems[8]. Understanding nanoparticle biodistribution *in vivo* is crucial for these applications[9]. Many techniques can measure the total accumulation of inorganic materials in tissues, but few methods trace inorganic particles at the single-cell level[10,11]. Flow cytometry and confocal microscopy rely on fluorescence, however for nanoparticles that lack intrinsic fluorescence, a suitable fluorophore must be attached. This introduces several sources of error, due to label degradation, dissociation from NPs, and altered *in vivo* behaviour.

Label-free approaches for detection of particles such as electron microscopy and tomography suffer from low throughput[12–14]. Laser ablation inductively coupled plasma mass spectrometry (LA-ICP-MS) enables quantitation of metal contents at the single-cell level with additional insights on sub-cellular localization of NPs, however this image-based method also suffers from low throughput (tens to hundreds of cells typically analysed) and relatively low sensitivity (requiring millions of atoms per cell)[15–17]. Single particle ICP-MS (SP-ICP-MS) is another ICP-based method that utilizes time-resolved mode to enable direct quantification of the number concentration, size distribution of NPs, and their state of agglomeration[18,19]. It has allowed for single-cell analysis of metal-containing cells when the cell concentration was carefully optimized to avoid overlapping cells at the detector[20,21]. However, SP-ICP-MS is only suitable for NPs larger than 20 nm in diameter and is usually coupled with other methods such as LA-ICP-MS to determine NP cellular distribution and quantitation[22]. Currently there are no label-free measurement techniques that can quantify inorganic nanomaterials of arbitrary size/chemistry in single cells at high throughput[11].

Mass cytometry is a recently developed method merging time-of-flight ICP-MS with flow cytometry[23]. Single-cell suspensions are labelled with metal isotope-tagged antibodies or other binding probes. Individual cells are then ionized in an argon plasma followed by time-of-flight mass spectrometry, which enumerates each metal species present in the resulting ion cloud[24,25]. Current Helios mass cytometry instruments permit up to 50 metal isotope labels (atomic weights ranging from 75 to 209) to be detected simultaneously on a single cell. Such highly multiparametric detection has offered new insights into the complexity of biology, in applications ranging from deep phenotyping of tumours to immune system signalling pathways[26,27].

Here we show for the first time that when combined with nanoparticle calibration, mass cytometry can also be used as a powerful fluorophore label-free method to track inorganic nanoparticles in tandem with highly multivariate cellular phenotyping, enabling quantitative analysis of the *in vivo* fate of inorganic nanomedicines. Using gold NPs (AuNPs) as a representative inorganic nanomaterial with relevance for diverse biomedical applications[6,7,28–32], we demonstrate the capacity of mass cytometry to enumerate nanoparticles in individual cells with a sensitivity orders of magnitude greater than flow cytometry. We show that mass cytometry overcomes challenges in fluorescence-based analysis of autofluorescent tissue cells, and illustrate the value of combined single cell NP detection with antibody-based phenotyping, using insights derived from mass cytometer analysis to select a nanoparticle composition that accumulates in dendritic cells for vaccination.

## Results

**AuNP per cell quantitation via mass cytometry.** We first synthesized AuNPs with comparable inorganic core diameters but three different surface chemistries expected to have distinct biodistributions and cellular uptake *in vivo* (Fig. 1a): 3-mercapto-1-propanesulfonate (MPSA) NPs, coated by a dense layer of short sulfonate-terminated ligands that strongly interact with water; 11-mercapto-1-undecanesulfonate/1-octanethiol (MUS/OT) NPs bearing an amphiphilic mixed ligand shell, which are water soluble but strongly interact with cell membranes;[33,34] and poly(ethylene glycol) NPs sterically stabilized by PEG to reduce opsonization by serum components[35]. The particles were relatively monodispersed with similar mean gold core diameters 2.5–4 nm and negative zeta potentials (Fig. 1b,c and Supplementary Table 1).

Pilot experiments established that gold was readily detected by mass cytometry analysis of cultured cells incubated with AuNPs using either CyTOF2 or Helios instruments. We first compared the sensitivity of mass cytometry and flow cytometry for detecting NP uptake, incubating BODIPY-labelled MUS/OT NPs[36,37] with RAW macrophages for 6 h, followed by flow cytometry or mass cytometry. Calibration of the TOF detector (see Methods) enabled a direct enumeration of gold ions, and thereby mean numbers of nanoparticles accumulated per cell. Gold uptake by macrophages was clearly detectable by mass cytometry across this entire concentration range (with detector saturation occurring at an upper detection limit of $\sim 1.5 \times 10^6$ particles per cell, Fig. 2c), whereas NPs at concentrations of 0.1 µg ml$^{-1}$ or lower were not detected in cells using flow cytometry (Fig. 2a,b). Using the bulk analysis method of inductively coupled plasma atomic emission spectrometry[10] (ICP-AES) as an independent measure, we found that the mass cytometer-determined count of AuNPs per cell (averaged from 16,000 cells) was in close agreement with the average gold content calculated from ICP-AES analysis of $10^7$ cells (Fig. 2c). The lower limit of detection using the Helios mass cytometer was first calculated as three times the standard error of regression for the best fit to the dual counts versus dilution data (Supplementary Fig. 1), which resulted in a detection limit of $\sim 4.2$ NPs per cell. However, the first particle concentration to be statistically significant was $\sim 10$ NPs per cell ($8 \pm 2$ NP per cell at 0.005 µg ml$^{-1}$ incubation concentration), a dosage that could only be detected in ICP-AES using a $10^4$-fold greater number of cells. Overall, mass cytometry was $\sim 2,400$ times more sensitive than flow cytometry in detecting 3 nm BODIPY-labelled MUS/OT AuNP uptake, and provided a direct quantification of total particles per cell.

**Label-free NP quantitation and cellular phenotyping *in vivo*.** We next compared CyTOF2 and flow cytometry for analysis of AuNPs taken up by cells *in vivo*. BODIPY-labelled MUS/OT particles, which we have previously shown exhibit cell penetrating properties by dispersing through cell membranes[33,34,36], were administered intratracheally into the lungs of mice. Lung tissues were collected 2 h later, stained with antibodies to CD326, and then analysed by the two methods in parallel. A significant autofluorescence signal from the tissue cells was observed in the BODIPY channel, a common issue in flow cytometry (Fig. 3a). However, AuNP uptake was clearly detected in a fraction of both epithelial cells (CD326$^+$) and CD326$^-$ cells, accounting for $\sim 13\%$ of all lung cells (black gates in Fig. 3a,b). By contrast, CyTOF2 analysis revealed that by 2 h MUS/OT particles were detectable in virtually all of the cells in NP-dosed lungs (Fig. 3a). While distinct AuNP$^{hi}$ CD326$^+$ and CD326$^-$ populations were observed corresponding to the AuNP$^+$ populations detected by flow cytometry (black gates, Fig. 3a), the majority of the

remaining epithelial and other lung cells were also clearly positive for MUS/OT particles (Fig. 3a,b). We confirmed that this result was not caused by Au retention in the instrument by analyzing untreated cells (gold-negative cells) before and after gold-containing cells, and found near-zero dual counts in both gold-negative cell samples. To verify that this discrepancy was due to a failure of flow cytometry to detect low level BODIPY-NP signals above background cellular autofluorescence, we flow-sorted $5 \times 10^6$ CD326$^-$BODIPY$^-$ cells from lung tissues (red gate, Fig. 3a) and analysed their gold content via conventional ICP-AES. The AuNP level in this cell population was non-trivial—38,000 particles per cell on average (Fig. 3c)—a value that was not statistically different from the mean AuNP content determined by CyTOF2 in the CD326$^-$AuNP$^{lo}$ population (red gates in Fig. 3a,d).

We next intratracheally administered a low dose of MUS/OT NPs (1 µg), recovered lung tissues 24 h later, and stained with nine different metal-chelated antibodies to leukocyte cell surface markers for mass cytometry analysis. Gating separately 'Au low' versus 'Au high' cells (Fig. 3e), CyTOF2 revealed a CD45$^+$CD11b$^-$ lymphocyte population present only among the 'Au low' cells, which included AuNP$^+$ B-cells, CD4$^+$ T-cells and CD8$^+$ T-cells (Fig. 3f–h). Alveolar macrophages (AMs), an important target for antimicrobial drug delivery[38], were located in the 'Au high' population (Fig. 3i–k), and these cells contained 8-fold more nanoparticles than dendritic cells (DCs) and 18-fold more gold than B/T-cells (Fig. 3l). Notably, at 24 h no BODIPY signal was detectable in any cell population by flow cytometry, suggesting either degradation or loss of the fluorophore by this time point. Thus, multiple issues associated with fluorescence detection of nanoparticles can be overcome through mass cytometer analysis.

**Mass cytometry data-guided therapeutic development**. We finally tested the utility of mass cytometry for guiding the design of novel AuNP-based therapeutics. Bulk ICP-AES analysis of excised tissues showed that subcutaneous injection of MUS/OT NPs resulted in striking accumulation in draining inguinal and axillary lymph nodes (LNs), 13-fold higher than PEG NPs (Fig. 4a). To evaluate the cellular biodistribution of these particles, we carried out mass cytometry analysis of LNs. CyTOF detected MUS/OT particles in B220$^+$ B-cells, CD4$^+$ and CD8$^+$ T-cells, CD11b$^{+/-}$CD11c$^+$ dendritic cells, as well as neutrophils and F4/80$^+$ macrophages (Fig. 4b). However, the greatest particle accumulation ($\sim$2-fold greater than CD11b$^-$CD11c$^+$ DCs or T-cells) was detected in CD11b$^+$CD11c$^+$ myeloid dendritic cells (Fig. 4c). Both PEG NPs and MPSA NPs showed much lower accumulation in all cell types analysed (Fig. 4c). The preferential accumulation of MUS/OT particles in myeloid DCs revealed by mass cytometry prompted us to test these particles for vaccine delivery. A fluorophore-labelled peptide antigen derived from ovalbumin (SIINFEKL) was conjugated to MUS/OT particles through an alkanethiol linker, providing $\sim$9 peptides per particle (Fig. 4d and Supplementary Fig. 2). C57Bl/6 mice were then vaccinated with free peptide or peptide-MUS/OT NPs mixed with CpG DNA (as adjuvant). As shown in Fig. 4e,f, MUS/OT-mediated peptide delivery greatly increased the potency of the peptide vaccination, eliciting at peak $\sim$6-fold more CD8$^+$ T-cells than the equivalent dose of free SIINFEKL peptide, and greater than a 5-fold higher dose of free peptide or immunization with free FITC-SIINFEKL-linker construct (Fig. 4f). MUS/OT-peptide-vaccinated mice challenged with ovalbumin-expressing B16F10 melanoma tumour cells at day 150 exhibited robust cytokine-producing CD8$^+$ T-cell responses, and these animals were fully protected from tumour outgrowth,

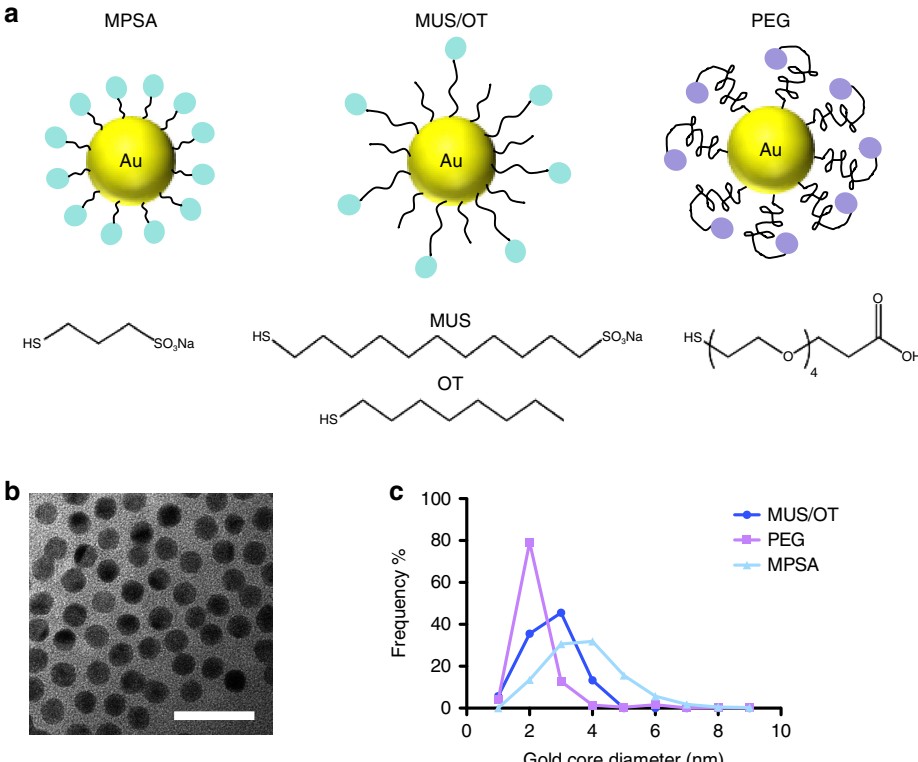

**Figure 1 | Gold nanoparticle ligand chemistry and size distribution.** (**a**) Schematics of MPSA (3-mercapto-1-propanesulfonate) coated AuNPs, MUS (11-mercapto-1-undecanesulphonate) and OT (1-octanethiol) mixed ligand-coated AuNPs, and PEG (tetraethylene glycol)-coated AuNPs. (**b**) Representative TEM image of MUS/OT NPs (scale bar 10 nm). (**c**) Size distributions of MPSA, MUS/OT and PEG NPs determined from TEM.

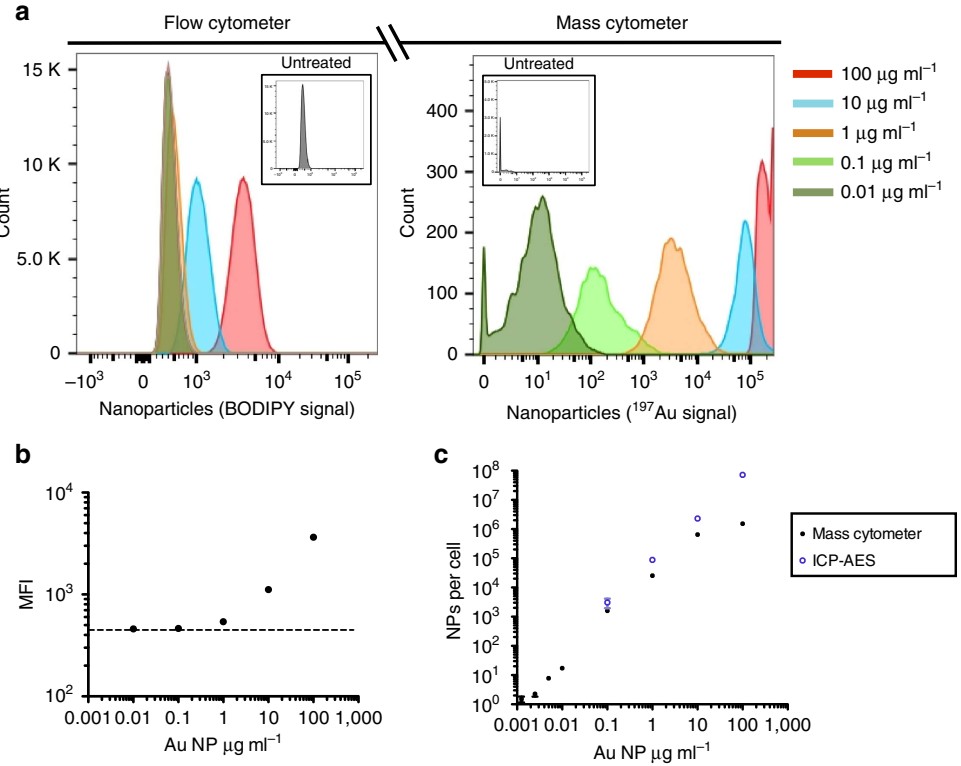

**Figure 2 | Sensitive detection of AuNPs in single cells with a wide dynamic range.** RAW macrophages were incubated with BODIPY-labelled MUS/OT NPs at 100, 10, 1, 0.1, 0.01 μg ml$^{-1}$, or left untreated for 6 h at 37 °C, washed three times, then analysed by mass cytometry or flow cytometry ($n = 3$ samples per group). (**a**) Histogram of AuNP levels detected in cells treated at five different concentrations. (**b**) Median fluorescence intensity (MFI) of cells treated at five different concentrations as assessed by flow cytometry ($n = 3$ samples per group). (**c**) Quantitative analysis of number of AuNPs per cell at five different treatment concentrations acquired by mass cytometry compared with parallel bulk measurements of AuNP uptake by ICP-AES. Shown are mean ± s.d. determined from triplicate samples.

in contrast to free peptide-immunized controls (Fig. 4g–i and Supplementary Fig. 3). While much remains to be done to fully understand the mechanisms, this example illustrates the power of single-cell inorganic NP analysis coupled with multiparameter phenotyping to develop novel nanomedicines.

## Discussion

Inorganic nanoparticles are being designed for diverse biomedical applications[1–8]. A key issue for any novel nanomedicine is characterization of the fate of the materials *in vivo*, at the tissue and cellular levels. Fluorescence-based methods such as confocal microscopy and flow cytometry are well established in tracking nanomaterial biodistributions at the single-cell level[10,11]. However, for nanomaterials that do not intrinsically fluoresce, achieving stable association of dyes with the particles *in vivo* is a significant challenge. Surface-functionalized labelling molecules may degrade or disassociate from nanomaterials, decreasing the intensity, reliability and accuracy of biodistribution outcomes. Methods that directly detect nanoparticle core atoms may overcome the above-mentioned technical issues. Here we demonstrated that mass cytometry could be used to simultaneously provide a quantitative measurement of nanoparticle uptake on thousands of single cells, together with measurement of expression levels of a large panel of cellular proteins provided by antibody-based markers that provided detailed identification of each cell analysed. Mass cytometry was 2,400-fold more sensitive than fluorescence labelling/flow cytometric detection of gold nanoparticle uptake in cells *in vitro*, and *in vivo*, this method provided sensitive detection of

nanoparticles in conditions where tissue autofluorescence and dye loss made traditional fluorescence-based tracking impossible.

Analytical tools are most powerful when used in combination[39], and we illustrated this here by analysing how the surface chemistry of gold nanoparticles impacted the tissue- and cell-level biodistributions of gold nanoparticles. Using mass cytometry, which provides detailed single-cell level analysis, together with ICP-AES, which can readily provide quantitative measurements of total inorganic nanomaterial content in a tissue, we analysed the biodistribution of three types of gold nanoparticles with distinct organic surface ligands. This combined analysis identified amphiphilic MUS/OT ligand compositions that led to very high total lymph node accumulation (at the tissue level) and preferential uptake in myeloid dendritic cells (at the cellular level). This prompted us to evaluate these amph-NPs as a platform for enhanced vaccine delivery. We showed that amph-NPs drastically improved peptide vaccine responses and were effective in protecting against tumour outgrowth.

This paper provides the first proof of concept demonstration that using mass cytometry, a fast, accurate, high-sensitivity screening of suitable inorganic nanoparticles for a particular application can be performed in a high-throughput manner. Compared with LA-ICP-MS scanning speeds of ∼8 μm per sec (∼1 cell per second)[15] and SC-ICP-MS analysis at ∼3 cells per second, mass cytometry offers detection speeds of ∼2,000 events per second. Thus, in a 3.5 h typical experiment (including the time for tissue isolation, cellular staining and analysis) 900,000 cells can be readily analysed at the single-cell level. This method should be applicable to the sensitive detection of many

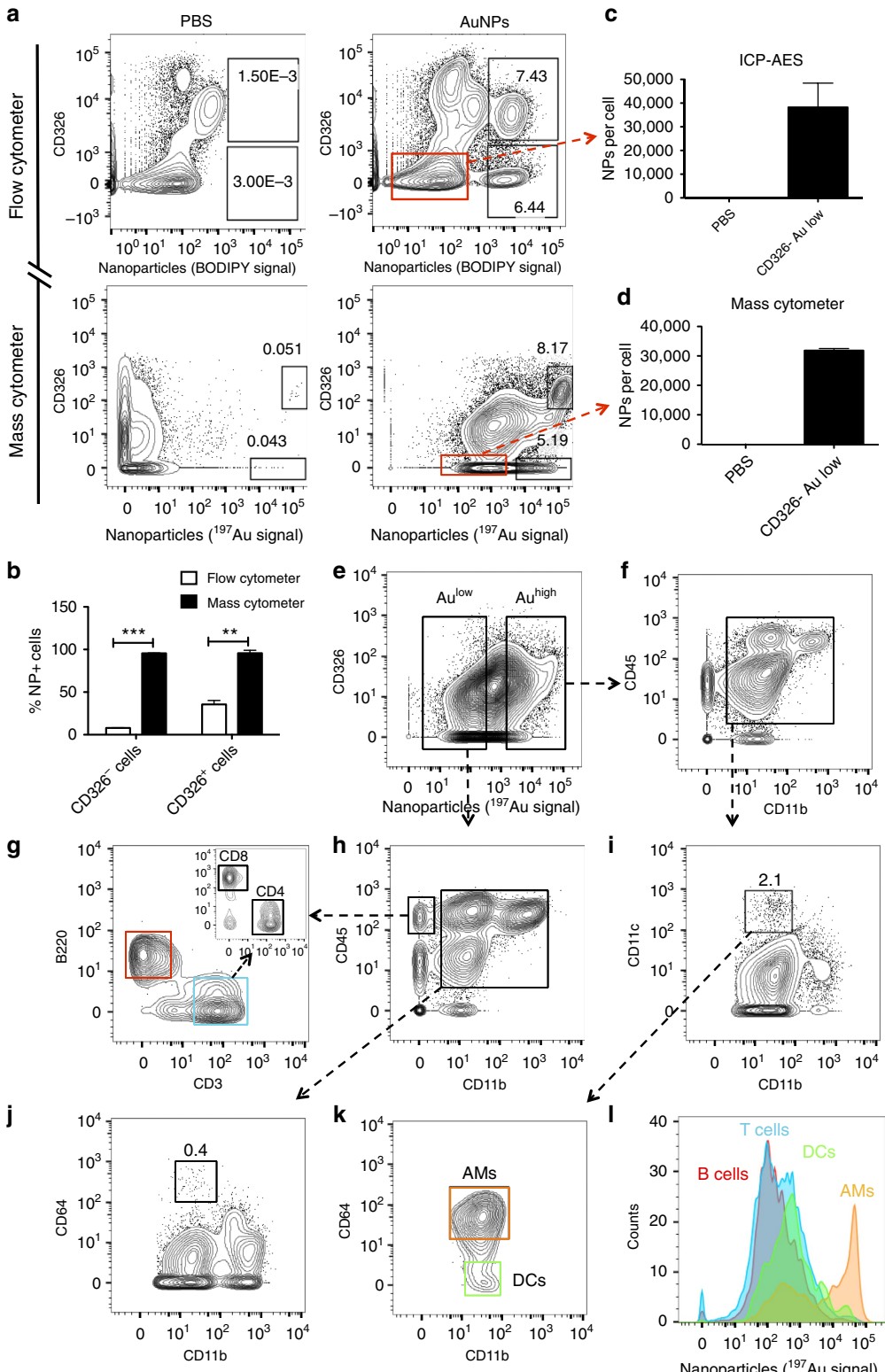

**Figure 3 | AuNPs detected by mass cytometer in all lung cell populations.** (**a–d**) BODIPY-MUS/OT AuNPs (50 μg in PBS) or saline were administered intratracheally (i.t.) to C57Bl/6 mice (*n* = 4 per group, pooled from two separate experiments). Two hours post administration, lung tissues were digested and stained with labelled antibodies followed by mass cytometry or flow cytometry analysis. (**a**) Parallel mass cytometry and flow cytometry analysis of lung cells stained with metal ($^{165}$Ho)-chelated or FITC-labelled CD326 antibodies. (**b**) Mean percentages of epithelial and non-epithelial cells detected as AuNP-positive by flow cytometer versus mass cytometer analysis. (**c**) Five million cells defined as CD326$^-$BODIPY$^-$ in flow cytometry (red gate in **a** upper panel) for AuNP-treated or saline control lung tissues were sorted for bulk ICP-AES quantification of AuNPs per cell. (**d**) Number of NPs per cell determined from direct Au analysis in the mass cytometry CD326$^-$ Au$^{lo}$ population (red gate in **a** lower panel). (**e–k**) Representative mass cytometry gating analysis of cells isolated from lungs 24 h post i.t. injection of 1 μg MUS/OT NPs. (**l**) Histogram of gold intensity in B-cells (CD45$^+$ B220$^+$ CD3$^-$), T-cells (CD45$^+$B220$^-$CD3$^+$), dendritic cells (DCs, CD45$^+$CD11c$^{hi}$CD11b$^+$CD64$^-$) and alveolar macrophages (AMs, CD45$^+$CD11c$^{hi}$ CD11b$^+$ CD64$^+$). Shown are means ± s.d. *P < 0.05; **P < 0.01, ***P < 0.001 by unpaired *t*-tests.

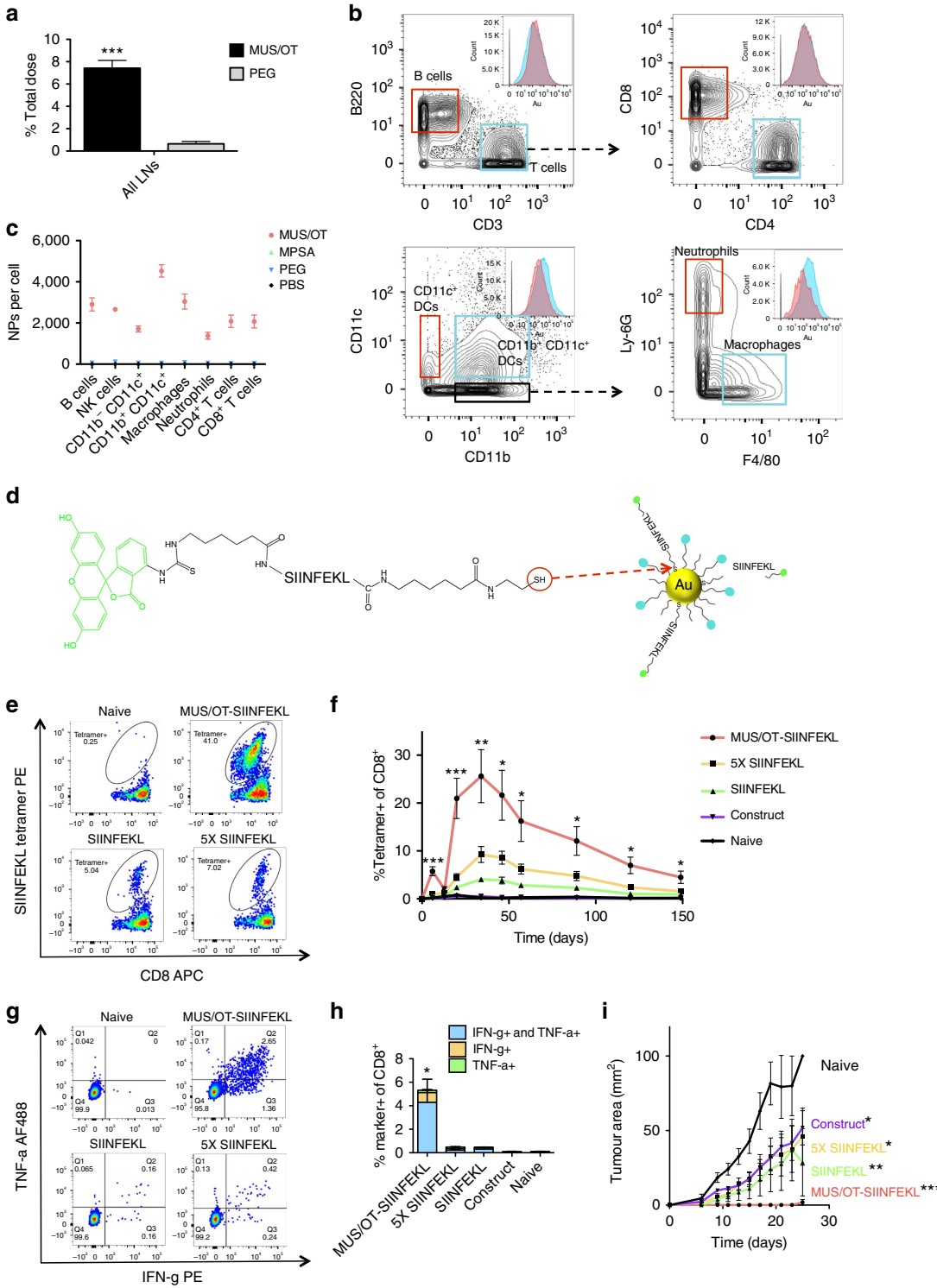

**Figure 4 | Dendritic cell uptake of MUS/OT NPs correlates with effective vaccine response.** (**a**) C57Bl/6 mice (*n* = 5 mice per group) were injected s.c. with 300 μg MUS/OT or PEG AuNPs, followed by ICP-AES quantification of total gold NP accumulation in lumbar, inguinal, axillary LNs 24 h later. Shown are mean ± s.d. ***P < 0.001 by unpaired *t*-tests. (**b,c**) C57Bl/6 mice (*n* = 3 per group) were injected s.c. with 100 μg MUS/OT, MPSA or PEG AuNPs, and lymph nodes were excised, digested and stained for CyTOF2 analysis 24 h later. Shown are representative Au histograms showing AuNP levels in various leukocyte populations (**b**) and mean number of NPs per cell for lymph node cell populations (**c**). (**d**) Schematic structure of SIINFEKL peptide construct and coupling to MUS/OT NP surface. (**e–i**) C57Bl/6 mice (*n* = 5 per group) were immunized s.c. on days 1 and 14 with 8 μg CpG mixed with SIINFEKL-conjugated MUS/OT NPs (10 μg peptide), 50 μg SIINFEKL peptide, 10 μg SIINFEKL peptide or 10 μg SIINFEKL peptide construct. Animals were then challenged with 2.5 × 10^5 B16-OVA tumour cells s.c. in the flank on day 150. Shown are representative flow cytometry plots of SIINFEKL tetramer staining (**e**) and mean SIINFEKL tetramer+ CD8+ T cells in blood on day 29 (**f**), representative intracellular cytokine staining flow cytometry (**g**) and mean percentages of cytokine+ T cells (**h**) 6 days after tumour challenge, and tumour size measurements over time (**i**). Shown are means ± s.e.m. *P < 0.05; **P < 0.01, ***P < 0.001 by one-way ANOVA with Bonferroni post tests.

other inorganic nanomaterials ranging from 75 to 209 a.m.u., including elements already used in biomedical nanoparticles such as platinum[3], bismuth[4], gadolinium[5], palladium[8] and lanthanides[40]. Exceptions will likely be elements that have high endogenous concentrations *in vivo*, such as molybdenum[41] and selenium[42], though the mass range of current mass cytometry instruments was explicitly designed to exclude elements prevalent *in vivo* (generally <75 a.m.u.), which would otherwise saturate the TOF detector. An important consideration is the maximum number of metal atoms that can be present in a single cell without saturating the detector. In the case of gold as studied here, this limit was approximately $1.3 \times 10^9$ gold atoms per cell, corresponding to $\sim 1.5 \times 10^6$ nanoparticles 3 nm in diameter, but would equate to 42,906 particles 10 nm in diam. or 343 particles 50 nm in diam. Thus, the dynamic range in particle enumeration will be highest for the smallest particle sizes. These limitations still make the method quite valuable since gold particles in biomedical applications are most commonly ≤50 nm in diameter.

In conclusion, single-cell mass cytometry by time of flight allows sensitive quantification of inorganic nanoparticle biodistributions in conjunction with highly multivariate phenotypic analysis. A limitation of this approach is the inability of mass cytometry to distinguish the precise physical state of nanomaterials (for example, aggregation state); like any other measurement approach complementary analytical techniques should be employed to obtain a complete picture of nanomaterials' fate *in vivo*. However, the ability to track the cellular distribution of diverse inorganic nanomaterials will facilitate our understanding of nanomaterial toxicology[9,43] and the development of new diagnostics and therapeutics.

## Methods

**Synthesis of gold nanoparticles.** MUS was synthesized following the previously reported methods[34]. All other chemicals were purchased from Sigma-Aldrich and used without further purification. 0.9 mmol gold(III) chloride trihydrate (99.9%) was dissolved in 150 ml of ethanol and 0.75 mmol of ligands (MUSOT: MUS and 1-octanethiol; MPSA: sodium 3-mercapto-1-propanesulfonate) with a desired molar ratio were added to the solution. After 15 min of stirring at 900 r.p.m., an ethanolic solution of sodium borohydride (NaBH₄—10 times molar excess of gold salt in 150 ml ethanol) was added dropwise to the previous solution at 25 °C. Black precipitates were almost immediately observed. The final solution was stirred for an additional 3 h to ensure reduction of the gold salts. The reaction was quenched by removing the solvent with centrifugation. To remove unreacted chemicals, additional washes with acetone and ethanol were carried out. Finally, water-soluble salts and any residual free ligands were removed using a centrifugal dialysis membrane (Amicon, MWCO 30 kDa). PEGylated gold nanoparticles coated by thiol-terminated poly(ethylene glycol; molecular weight 282.35 g mol⁻¹) were purchased from NanoPartz.

**Nanoparticle size and zeta potential characterization.** Nanoparticles in water were deposited on carbon-coated copper grids and images were acquired via JEOL 2010 FEG Analytical Electron Microscope (200 kV). Size distributions of AuNPs were characterized by ImageJ size analysis tools. For zeta-potential measurements carried out in the instrument Malvern Zetasizer NanoZS, NPs were dissolved in 10 mM NaCl solution, sonicated for 5 min and filtered through 0.20 μm syringe filters prior to measurements. The concentration of all nanoparticle solutions was 0.2 mg ml⁻¹. Error bars in zeta potential values represent three individual measurements of the same solution.

**Modification of gold nanoparticles with a fluorescent tag.** Fluorescent dye BODIPY 630/650-X NHS Ester (Invitrogen) and thiol linker (11-mercaptoundecyl amine hydrochloride; Prochimia, Poland) were used as received. 3 mg BODIPY dye and 1.5 mg thiol linker were dissolved in argon-purged amine-free dimethyl formamide and stirred for 6 h in the dark. 3 ml water was added to the solution and stored at 4 °C as a stock solution. To label the nanoparticles with thiol-functionalized Bodipy dye, 10 mg gold nanoparticles were dissolved in 0.75 ml of water in which 15 μl BODIPY stock solution was added. The reaction was left stirring for 48 h in the dark at 25 °C. Finally, 10 ml acetone was added to the reaction and NPs were washed at least three times to remove unreacted dye via centrifugation for 5 min at 14,000 r.p.m. in a tabletop centrifuge.

**Cell culture and *in vitro* treatment conditions.** RAW-Blue cells derived from RAW 264.7 macrophages were purchased from InvivoGen and cultured in DMEM-based cell culture media according to manufacturer's instructions. An ATCC mycoplasma testing PCR kit was used to ensure that all of the cells used in this study were mycoplasma free. BODIPY-conjugated MUSOT NPs were dissolved in cell culture media at 100 μg ml⁻¹ and diluted to various concentrations indicated in Fig. 2 and Supplementary Fig. 1. One and a half million cells per well were seeded overnight and the next day cells were treated with NP solution for 6 h at 37 °C with 5% CO₂. Six hours later, excess NP solution was removed and cells were washed in PBS three times. Cells were collected and split into three tubes for three separate analyses: flow cytometry, mass cytometry (using either a CyTOF2 or Helios mass cytometer instrument, both from Fluidigm), and ICP-AES.

**Cell isolation and antibody staining for mass cytometry.** C57BL/6 mice used in this study were 6–8 weeks old. All animal work was conducted under the approval of the Massachusetts Institute of Technology (MIT) Division of Comparative Medicine in accordance with federal, state and local guidelines. Cells from mouse lymph nodes were isolated by enzyme digestion method. Briefly, fresh enzyme mix was prepared by dissolving 0.8 mg ml⁻¹ of collagenase/dispase (Roche Diagnostics) and 0.1 mg ml⁻¹ DNase I (Roche Diagnostics) in RPMI-1640 medium. Each lymph node was pierced by a forcep and incubated in enzyme mix at 37 °C on a shaker for 30 min. Cells and tissue fragments in enzyme mix were mixed vigorously with a 1 ml syringe (without needle) for 30 s and quenched by adding 10 ml of ice-cold PBS with 1% BSA and centrifuged at 1,700 r.p.m. for 5 min. Cell pellets were resuspended in staining buffer followed by antibody staining and fixing: cells were incubated with a selected antibody cocktail (anti-mouse CD45 (30-F11)-147Sm; anti-mouse CD3e (145-2C11)-152Sm; anti-mouse CD8a (53-6.7)-168Er; anti-mouse CD4 (RM4-5)-172Yb; anti-mouse CD45R/B220(RA3B2)-176Yb; anti-mouse CD11b (M1/70)-148Nd; anti-mouse Ly-6G (Gr-1) (RB6-8C5)-174Yb; anti-mouse CD11c (N418)-142Nd; anti-mouse F4/80 (BM8)-159Tb; anti-mouse NK1.1 (PK136)-170Er; anti-mouse CD64 (X54-5/7.1)-151Eu; anti-mouse CD326 (EpCAM) (G8.8)-165Ho) at 25 °C for 30 min, excess antibodies were removed by centrifugation, and cells were stained with cell-ID Intercalator-Ir in fix and perm solution (detailed protocol available from Fluidigm website. https://www.fluidigm.com/productsupport/cytof-helios). Prior to analysis, fixed cells were washed in MaxPar staining buffer twice and MaxPar water once to remove excess iridium. Cells were resuspended at 0.5–1 million per ml in 1:10 calibration beads (EQ Four Element Calibration Beads, Fluidigm) in MaxPar water and 250–500 μl samples were analysed by a Fluidigm CyTOF2 at a flow rate of 0.045 ml min⁻¹ or Helios at a flow rate of 0.030 ml min⁻¹).

**Calculation of number of nanoparticles per cell.** The mean Au ion intensity in a cell population of interest measured by mass cytometry is termed the 'mean dual counts'. This value is proportional to the number of Au atoms per cell—and it is the product of the integral over time of detector intensity multiplied by the dual count coefficient of ¹⁹⁷Au. Conversion of dual counts to the number of gold atoms per cell was determined by a calibration using the transmission coefficient:

$$\text{Number of Au atoms per cell} = \frac{^{197}\text{Au mean dual counts}}{^{193}\text{Ir transmission factor}} \qquad (1)$$

The transmission coefficient for ¹⁹⁷Au cannot be directly measured in the cytometer, but can be measured for ¹⁹³Ir, which should have a degree of ionization very similar to gold (Ir and Au have ionization energies of 8.9760 and 9.2255 eV, respectively). The ¹⁹³Ir transmission coefficient was calculated as the dual counts of ¹⁹³Ir detected for the instrument tuning solution divided by the number of ¹⁹³Ir atoms introduced in the 0.25 p.p.b. Ir tuning solution (Fluidigm CAT#201072) determined as:

$$^{193}\text{Ir transmission factor} = \frac{^{193}\text{Ir dual counts}}{(\text{Ir atoms introduced}) \times (^{193}\text{Ir natural abundance})}$$
$$= \frac{^{193}\text{Ir dual counts}}{^{193}\text{Ir atoms introduced}} \qquad (2)$$

$^{193}\text{Ir atoms introduced}$

$$= \frac{(\text{Ir concentration})(\text{flow rate})(\text{integration time})(\text{natural abundance})(\text{Avogadros number})}{\text{Isotope mass}} \qquad (3)$$

Using the variables tuning solution Ir concentration ($2.5 \times 10^{-13}$ g μl⁻¹), flow rate (0.75 μl s⁻¹ for CyTOF2 or 0.5 μl s⁻¹ for Helios), the integration time (CyTOF2: 2.666 s; Helios: 4 s), the natural abundance of ¹⁹³Ir (0.627), Avogadro's number $6.02 \times 10^{23}$, and isotope mass (193 g mol⁻¹). The number of Au atoms per NP was calculated based on the assumption that AuNPs are monodispersed spheres with an FCC lattice structure (Au lattice constant = 0.40758 nm). As a result, a 2.8 nm AuNP contains $\sim 677$ atoms. Finally, the number of NPs per cell was calculated by the number of atoms per cell divided by the number of atoms per NP.

**Limit of detection analysis.** To assess the detection limit of 3 nm MUS/OT AuNPs per cell on the Helios instrument, RAWblue cells were treated with BODIPY-MUS/OT AuNPs at 0.1, 0.01, 0.005, 0.0025 and 0.00125 $\mu g\,ml^{-1}$ for 6 h at 37 °C in 10%FBS-containing DMEM. Cells were washed with PBS twice prior to fixation in the presence of Ir cell-ID (1:1,000) DNA stains. Cells were resuspended at 0.5 million per ml in 1:10 calibration beads (EQ Four Element Calibration Beads, Fluidigm) in MaxPar water and 200 $\mu l$ samples were analysed by a Fluidigm Helios at a flow rate of 0.5 $\mu l\,s^{-1}$. Events double positive for the $^{193}Ir$ and $^{191}Ir$ DNA stains were gated, and singlets (excluding debris and doublets) were gated using a $^{191}Ir$ versus event length plot. The mean number of particle per cell (determined from the entire population of both NP$^{+}$ and NP$^{-}$ cells) was calculated using the $^{193}Ir$ transmission factor as described in the previous method section. The mean ± s.d. number of particles per cell for each condition was determined by triplicate analysis of the same cell samples.

**Bulk measurement of AuNP in cells via ICP-AES.** Cells or tissue samples were dissolved in 1 ml freshly prepared aqua regia for 3 days to dissolve AuNPs. The solution was then diluted in 3–4 ml of 2% nitric acid immediately prior to ICP-AES analysis on a Horiba Activa.

**Peptide conjugation and quantification.** SIINFEKL peptide constructs were custom synthesized by LifeTein with the following structure: (N terminus) FITC-aminohexanoic acid (Ahx)-SIINFEKL-Ahx-cysteamide (C terminus), with purity > 95%. Lyophilized peptide was dissolved in DMF at 1 mg ml$^{-1}$. A mass ratio of gold:peptide of 4:1 in DMF was mixed in a glass vial and placed on a shaker to allow coupling reaction for 4 days. To remove uncoupled peptide, the MUS/OT-peptide solution was first diluted in water (<5% DMF) and spun at 3,500 r.p.m. for 15 min in an Amicon 10 kDa MWCO centrifugal tube. The above-mentioned washing step was performed repeatedly for a total of four times. To quantify peptide conjugation efficiency, 20 $\mu l$ beta-mercaptoethanol (14.3 M stock solution) and 20 $\mu L$ DMF were added to an aliquot (0.1 mg in 60 $\mu l$ H$_2$O) of purified MUSOT-peptide conjugates and allowed to react for 48 h on a shaker at 25 °C. Peptide conjugation efficiency was determined by fluorescence readout of FITC at excitation of 488 nm and emission of 520 nm using a standard curve made using uncoupled MUSOT particles doped with known amounts of peptide construct subjected to the same reaction conditions. The mass ratio of conjugated peptide to gold was determined to be ∼ 51 $\mu g$ peptide per mg gold, which corresponds to ∼9 peptide constructs per NP.

**Vaccine delivery and antigen-specific CD8$^{+}$ T cells.** Eight-week-old female C57BL/6 mice were immunized (primed on day 1, boosted on day 14) with 8 $\mu g$ of CpG (ODN 1826 VacciGrade, InvivoGen) mixed with SIINFEKL peptide (10 $\mu g$ peptide-conjugated AuNP, 10 $\mu g$ free peptide, 50 $\mu g$ free peptide or 10 $\mu g$ free peptide construct). Vaccines were formulated in 100 $\mu l$ sterile saline with half of the volume injected subcutaneously on either side of the tail base. To monitor antigen-specific T cells, mice were bled, and blood samples were processed as follows: 100 $\mu l$ of blood was incubated with 500 $\mu l$ ACK lysis buffer at 25 °C for 5 min followed by centrifugation, then this process was repeated for a second round of lysis. Cells were incubated in tetramer staining buffer (PBS, 1% BSA, 5 mM EDTA, 50 nM dasatinib), Fc block, and OVA tetramer (iTAg Tetramer/PE-H-2K$^b$ OVA, MBL) in the dark for 45 min at 25 °C. Anti-CD8a (53-6.7) APC antibody (1:200) was added to cell solutions and incubated for an additional 15 min at 4 °C. Cells were washed twice in flow cytometry buffer containing 100 nM DAPI, and run on a BD FACS LSR Fortessa. Data were analysed using FlowJo.

**Tumour cell culture and tumour inoculation in vivo.** B16-OVA cells were a kind gift from Dr Glenn Dranoff at the Dana–Farber Cancer Institute. B16-OVA cells were cultured in complete DMEM (DMEM supplemented with 10% FBS, 100 units per ml penicillin, 100 $\mu g\,ml^{-1}$ streptomycin and 4 mM L-alanyl-L-glutamine), maintained at 37 °C and 5% CO$_2$, and passaged when 70–80% confluent. A challenge of $2.5 \times 10^5$ B16-OVA cells was injected subcutaneously on the right flank of previously immunized mice in 50 $\mu l$ of sterile saline. Tumour size was measured (longest dimension × perpendicular dimension) three times weekly, and an area was calculated by multiplying these dimensions. Mice were killed when tumour area exceeded 100 mm$^2$. All animal work was conducted under the approval of the Massachusetts Institute of Technology (MIT) Division of Comparative Medicine in accordance with federal, state and local guidelines.

**Intracellular cytokine staining.** Peripheral blood mononuclear cells were isolated from immunized mice and cultured in RPMI supplemented with 10% FBS, 100 units per ml penicillin, 100 $\mu g\,ml^{-1}$ streptomycin and 4 mM L-alanyl-L-glutamine with 10 $\mu g\,ml^{-1}$ SIINFEKL peptide. After 2 h, Brefeldin A (1/1,000, eBiosciences) was added to inhibit cytokine secretion. After 6 h total incubation with peptide, cells were washed, stained extracellularly with anti-CD8a (53–6.7, eBioscience), fixed and permeabilized (BD Cytofix/Cytoperm), and stained intracellularly with anti-IFN-γ (XMG1.2, eBioscience) and anti-TNF-α (MP6-XT22, eBioscience). Cells were run on a BD FACS LSR Fortessa and data was analysed using FlowJo.

**Data availability.** The data that support the findings of this study are available from the corresponding author upon reasonable request.

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

## Acknowledgements

We thank Nicole E. Paul for technical assistance with CyTOF sample analysis at the Dana-Farber Cancer Institute, Boston, MA. We acknowledge the Center for Materials Science and Engineering (CMSE) at MIT for the use of TEM and ICP-AES facilities. This work was supported in part by the US Army Research Laboratory and the US Army Research Office through the Institute for Soldier Nanotechnologies, under contract number W911NF-13-D-0001 and the NIH (awards CA174795 and CA172164). We also acknowledge the EU Horizon2020 FutureNanoNeeds Project.

## Author contributions

Y.-S.S.Y. designed and performed most experiments, analysed the data and wrote the manuscript; P.U.A. designed and performed pilot vaccine studies; K.D.M. designed and performed vaccine and tumour studies, analysed the data and wrote the corresponding methods. K.R. assisted with i.t. injections; L.T. assisted with i.v. injections; Y.-S.S.Y. and A.B. synthesized and characterized nanoparticles. A.B. and F.S. provided MUS/OT nanoparticles and edited the manuscript; D.J.I. designed and supervised the research and wrote the manuscript.

## Additional information

**Competing financial interests:** The authors declare no competing financial interests.

**Publisher's note**: 

**DOI: 10.1038/ncomms15343**     **OPEN**

# Erratum: High-throughput quantitation of inorganic nanoparticle biodistribution at the single-cell level using mass cytometry

Yu-Sang Sabrina Yang, Prabhani U. Atukorale, Kelly D. Moynihan, Ahmet Bekdemir, Kavya Rakhra, Li Tang, Francesco Stellacci & Darrell J. Irvine

*Nature Communications* 8:14069 doi: 10.1038/ncomms14069 (2017); Published 17 Jan 2017; Updated 14 Aug 2017

In Fig. 4d of this Article, the schematic was incorrectly illustrated with one SIINFEKL peptide detached from a MUS/OT nanoparticle surface. The correct version of Fig. 4 appears below as Fig. 1.

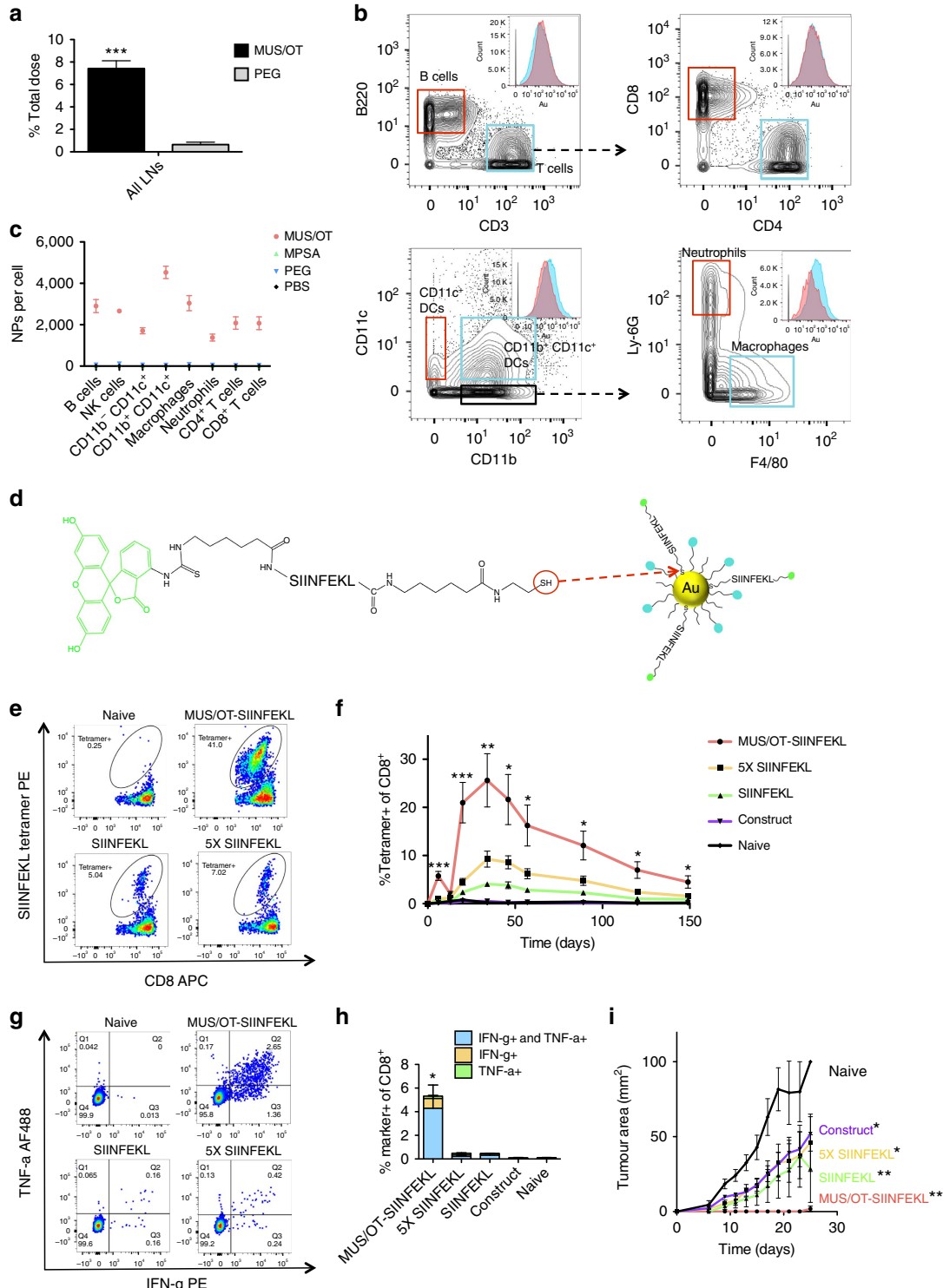

**Figure 1**

