## [Peer Review File · Nature Communications]

Reviewer #1 (Remarks to the Author):

This is an example of a creative use of new technology of mass cytometry for detection of inorganic nanoparticles, study of their uptake in different cell types, and use of the finding of the uptake study to efficiently deliver vaccine and affecting protection against tumor outgrowth. The work is original and should be of high interest to the Nature Communications audience. Suggest acceptance after revision.

1) Assessment of the number of nanoparticles per cell is based on the transmission factor for Iridium obtained by measuring CyTOF instrument response to tuning solution. If this is tuning solution which was supplied by DVS Sciences (now Fluidigm Canada) with the instrument, its iridium concentration is 0.25 ppb, not 0.5 ppb claimed by the authors (line 266 of the methods). It is also not clear why calculation of the transmission factor for ^{193}Ir requires division of the dual counts by ^{193}Ir natural abundance. CyTOF measures isotope-specific Ir, e.g. ^{193}Ir dual counts should be divided by the number of ^{193}Ir atoms introduced (per same time as measurement time) with the tuning solution. Perhaps the authors could provide more detail on this calculation: what aspiration flow was used, and what integration time, and what transmission factor for iridium was calculated as a result. Please also provide reference to the source of the tuning solution.

2) In Fig.2 c, standard error of mean is shown. If this is calculated as standard deviation divided by the square root of the number of events, the resulting error is artificially low (because of very large number of events). Showing actual standard deviation of the measured distributions would better represent the spread of the data.

3) it is claimed that the detection limit is 10 Au nanoparticles per cell. Is this calculated as per definition of the detection limit, e.g. using equivalent concentration of the 3 standard deviations of the background? Please provide information on how the detection limit was calculated.

4) Please provide correct model number for the instrument. As far as this referee aware, it is CyTOF2, which has higher sensitivity than CyTOF.

5) Generally speaking, the method is called mass cytometry, not CyTOF. CyTOF is the name of the first commercial instrument, and since then two more (CyTOF2, Helios) were introduced. Please include the reference to the original paper on mass cytometry as a reference (line 45): "Mass cytometry: technique for real time single cell multitarget immunoassay based on inductively coupled plasma time-of-flight mass spectrometry", Analytical Chemistry 2009. Please use CyTOF in a context of the instrument name, and mass cytometry in a context of the method discussion.

6) References 15 and 16 are cited inaccurately: in both the mass range of CyTOF was from 103 to 193. CyTOF 2 has mass range of 89 -209. Current Helios, a CyTOF system, has the range which is given by the authors: 75 to 209, however, it was introduced only in 2015, while ref 15 and 16 are of 2012 and 2011, respectively.

7) In methods, the particulars of the instrumentation should be given (e.g. model, manufacturer). The fact that CyTOF2 model was used needs to be evident.

8) Relevant to the transmission factor discussion: one MUS/OT nanoparticle of 2.8 nm diameter can be calculated to have 670 atoms of gold. 10 such nanoparticles (claimed to be the detection limit) would have 6,700 atoms. Given 50 % ionization degree of gold in inductively coupled plasma, 3,350 Au ions would be generated from a cell containing 10 nanoparticles. This is just enough to generate one count at CyTOF2 detector. Figure 1 a of the supplemental information shows, however, that 1500 nanoparticles per cell generate mean raw signal (Au ^{197}Di , which is dual counts at the detector) of close to 1500, e.g. one count per one nanoparticle. There is a factor of ~ 10 discrepancy here. Is the horizontal axis showing dual counts, or the number of nanoparticles? Could you please also propose explanation (or suggest that there is no clarity yet, since this is a communication) why the distributions for 5,8,23 and 66 nanoparticles per cell are bimodal?

9) The abstract would benefit from inclusion of the vaccine delivery experiment result (tumor outgrowth control).

Impressive work, keep up!

Reviewer #2 (Remarks to the Author):

The paper by Yang et al studied 3 nm gold nanoparticles and their interaction with cells. Initially, they used BODIPY-labeled gold NP that were taken up nonspecifically by cells, and demonstrated that the amount of gold could be accurately measured by CyTOF. Similarly, the signal was in proportion to the concentration of NP used; the use of ICP-AES helped confirm proportionality.

Later, the authors extended the work to show that the NPs were taken up to different extents by cells present in lymph node tissues. They demonstrated that myeloid dendritic cells accumulated the most. Injecting antigen-conjugated NPs into mice allowed much greater survival against tumor challenge than free antigen or linker-antigen. This could be of great use as a general method of vaccination/antigen-presentation in the future: it would have the advantage of being a well-characterized particle to which virtually any antigen could be conjugated.

Nanoparticle conjugations and their introduction into animals or cells isn't new, but I do feel that the authors did a good job of characterizing their construct, and demonstrating effectiveness in the tumor challenge model. Their particles are also "visible" by CyTOF, which could help multiparameter characterization of other, future constructs.

Major issues:

1. There are no calculations stating the approximate number of Au ions per particle. This has great importance for the CyTOF: the detector can be saturated, which is a likely reason for the CyTOF line deviating from the ICP-AES line in Fig 2c, and the lower-than-expected signal in Fig 2a for 100 ug/mL.

2. I am not convinced that their technique can actually accurately, reproducibly detect down to 10 particles/cell as stated on Line 87. Since the authors used a cell line (rather than a mixture of cells like PBMCs) for experiments like Fig 2 and Suppl Fig 1, I would expect a consistent, unimodal cell labeling.

Looking at Fig 2a, the 0.01 ug/mL curve in dark green is not unimodal; there is a population with zero Au signal. The 0.1 ug/mL - 10 ug/mL curves are unimodal; the 100 ug/mL probably is too, but appears to be saturating the detector. Looking at Suppl Fig 1, the lowest concentrations (0.00125-0.01 ug/mL; 5.33-65.63 NP/cell) are completely bimodal.

I will agree that they can properly, reproducibly detect down to 0.1 ug/mL, but that corresponds to 1515 NP/cell, not 10 NP/cell. This is still substantially better (15-20x?) than the fluorescence data in Fig 2a, so I don't see this as any kind of failure.

Minor issues:

1. On page 1, lines 37-39, the authors talk about how "most nanoparticles lack intrinsic fluorescence". However, Qdots (particularly Cd-based ones) have been used in fluorescence flow and microscopy for decades. Also, the Bendall et al reference 16 used a CD3-Qdot in their CyTOF experiment.

Similarly, recent work detailing how different-sized gold nanoparticles can have intrinsic color different than that of bulk gold metal due to surface plasmon resonance and such seems relevant. This seems especially important, since several are commercially-available from Sigma, including some in the 5nm range that would be relevant to the 3nm particles detailed in the article.

2. In the caption for Fig 1c, the authors state "Size distributions and zeta potentials.... determined from TEM and dynamic light scattering, respectively". DLS doesn't give zeta potential. The Zetasizer NanoZS used by the authors is capable of doing both DLS and zeta potential, but they aren't the same: according to the manufacturer's website, the DLS is done by non-invasive back-scatter, while laser doppler micro-electrophoresis with PALS is used for zeta potential.

Aggregation state mentioned in line 186 should be measurable, at least in a general sense, by DLS looking at size distributions.

3. While Au197 is the only naturally-occurring, stable gold isotope, I would still recommend stating the mass measured in places such as Fig 2a right plot.

Reviewer #3 (Remarks to the Author):

This manuscript demonstrates the use of CyTOF technology and metal-tagged antibodies to determine the cellular fate of nanoparticles in vivo. The manuscript is not the first to report quantification of nanomaterials in single cells using ICP-MS based techniques (see e.g. Zheng, Talanta, 2013, 116:782), nor is it the first to co-detect nanoparticles and metal-tagged antibodies (see e.g. Managh, J. Immunol., 2014, 193:2600). However, the highly multiplexed nature of the work presented here (13 parameters) represents an advance over the existing literature, particularly in relation to the ability to quantify and correlate nanoparticle uptake with cellular phenotype. As is demonstrated for vaccine delivery, the method has the potential to influence the development of inorganic nanomedicines and thus will be of interest to workers in this field.

Overall the manuscript is presented in a clear and concise format, the data is well presented and the main conclusions are adequately justified by the data.

Specific points to be addressed:

1) Abstract: The authors claim that the technique is "applicable to diverse types of inorganic nanomaterials", yet the discussion within the paper does not provide sufficient information to support this statement. For instance, could the technique be applied to elements that have a higher background in vivo than Au? The authors should also comment on the mass range of the instrument (75-209 amu) in relation to this.

2) Introduction: The introduction should make reference to previous work in the field of single particle and single cell ICP-MS, and its relationship to detecting nanomaterials within cells using the CyTOF.

3) Page 2 Line 54: The term "label-free method" (used throughout the manuscript) could potentially be misleading, given that antibody based labels are subsequently used to identify the cellular destination of the nanoparticles. Consider rephrasing.

4) Page 2 Line 81: Was "the upper detection limit of $\sim 1 \times 10^6$ particles/cell" imposed by detector saturation? This warrants further discussion. As the upper limit is a function of the total metal content per cell, what might this mean for the detection of larger i.d. nanoparticles than the relatively small (~ 3 nm) ones studied here? Note that the tail of the 100 μ g/ml peak in Figure 2a appears to have been cut off. Were these cells not detected due to detector limitations?

- 5) Page 3 Line 102: Gold is known as a highly retentive element in the CyTOF system. Can the authors be sure that detection of gold in "virtually all of the cells" was not due to memory effects?
- 6) Page 5 Line 181: A comment on the total analysis time (and time taken for cellular staining), in comparison to the techniques mentioned in the introduction, would be helpful.
- 7) Figure 1: The label (d) is in the figure, but not referred to in the caption.
- 8) Figure 2c: The error bars mentioned in the legend are not visible on this figure. If the s.e.m. is too small to be observed, another measure e.g. s.d. might be more easy to visualise.
- 9) Page 7 Line 260: Despite the section title, it is not clear how the number of Au atoms per cell is converted to the number of gold nanoparticles per cell. Either the formula used should be included or an appropriate reference provided. An estimate of the uncertainty in the conversion should also be included somewhere in the manuscript.

Reviewer #1 (Remarks to the Author):

This is an example of a creative use of new technology of mass cytometry for detection of inorganic nanoparticles, study of their uptake in different cell types, and use of the finding of the uptake study to efficiently deliver vaccine and affecting protection against tumor outgrowth. The work is original and should be of high interest to the Nature Communications audience. Suggest acceptance after revision.

1) Assessment of the number of nanoparticles per cell is based on the transmission factor for Iridium obtained by measuring CyTOF instrument response to tuning solution. If this is tuning solution which was supplied by DVS Sciences (now Fluidigm Canada) with the instrument, it's iridium concentration is 0.25 ppb, not 0.5 ppb claimed by the authors (line 266 of the methods). It is also not clear why calculation of the transmission factor for ¹⁹³Ir requires division of the dual counts by Ir¹⁹³ natural abundance. CyTOF measures isotope-specific Ir, e.g. Ir¹⁹³ dual counts should be divided by the number of Ir¹⁹³ atoms introduced (per same time as measurement time) with the tuning solution. Perhaps the authors could provide more detail on this calculation: what aspiration flow was used, and what integration time, and what transmission factor for iridium was calculated as a result. Please also provide reference to the source of the tuning solution.

The Ir tuning solution is from Fluidigm Cat# 201072. The concentration is indeed 0.25 ppb and we have corrected this typo in the methods, and more explicitly presented the calculation for the transmission factor and gold ions/cell in the revised methods.

2) In Fig.2 c, standard error of mean is shown. If this is calculated as standard deviation divided by the square root of the number of events, the resulting error is artificially low (because of very large number of events). Showing actual standard deviation of the measured distributions would better represent the spread of the data.

We have changed the figure to display standard deviation instead of s.e.m. We have also updated Figure 2c with a correction to the ICP-AES values, as we discovered a minor error in accounting for cell dilutions in those measurements.

3) it is claimed that the detection limit is 10 Au nanoparticles per cell. Is this

calculated as per definition of the detection limit, e.g. using equivalent concentration of the 3 standard deviations of the background? Please provide information on how the detection limit was calculated.

The raw mean \pm s.d. dual counts of ^{197}Au for untreated cells was 0.163 ± 0.146 . By classic definition of the detection limit = mean + 3σ , the detection limit would be 0.601 dual counts, which was calculated to be 743 atoms (~ 0.89 AuNPs of 3 nm diam.) based on the calibrating Ir transmission coefficient measured in that experiment. However, in Supplementary Figure 1c, we showed that ~ 1.5 NPs ($0.00125 \mu\text{g/mL}$) per cell was not statistically significant compared to untreated cells. The standard error of regression (sy/x) shown in Supplementary Figure 1c was calculated to be 1.678. A commonly used definition of detection limit = $3sy/x + \text{intercept} = 3 * 1.678 - 0.82 = 4.21$ NPs per cell. However, the first data point to be statistically significant was 8 ± 2 NP per cell ($0.005 \mu\text{g/mL}$), therefore, we state that the detection limit is ~ 10 AuNPs (3 nm in diam.) per cell. We have updated the Methods to make this calculation clearer and comment more explicitly on the detection limit calculation in the main text on p. 3.

4) Please provide correct model number for the instrument. As far as this referee aware, it is CyTOF2, which has higher sensitivity than CyTOF.

All experiments were conducted using a CyTOF2 instrument, with the exception of the RAW-blue *in vitro* experiments for the determination of lower detection limits, which were measured using a Helios instrument (supplementary figure1). The instruments are now explicitly noted in the Methods.

5) Generally speaking, the method is called mass cytometry, not CyTOF. CyTOF is the name of the first commercial instrument, and since then two more (CyTOF2, Helios) were introduced. Please include the reference to the original paper on mass cytometry as a reference (line 45): "Mass cytometry: technique for real time single cell multitarget immunoassay based on inductively coupled plasma time-of-flight mass spectrometry", Analytical Chemistry 2009. Please use CyTOF in a context of the instrument name, and mass cytometry in a context of the method discussion.

We have included this reference in line 45 (now line 61) and corrected the usage of CyTOF vs. mass cytometry throughout the manuscript.

6) References 15 and 16 are cited inaccurately: in both the mass range of CyTOF was from 103 to 193. CyTOF 2 has mass range of 89 -209. Current

Helios, a CyTOF system, has the range which is given by the authors: 75 to 209, however, it was introduced only in 2015, while ref 15 and 16 are of 2012 and 2011, respectively.

We have corrected these errors.

7) In methods, the particulars of the instrumentation should be given (e.g. model, manufacturer). The fact that CyTOF2 model was used needs to be evident.

We have updated the Methods to make the model number evident.

8) Relevant to the transmission factor discussion: one MUS/OT nanoparticle of 2.8 nm diameter can be calculated to have 670 atoms of gold. 10 such nanoparticles (claimed to be the detection limit) would have 6,700 atoms. Given 50 % ionization degree of gold in inductively coupled plasma, 3,350 Au ions would be generated from a cell containing 10 nanoparticles. This is just enough to generate one count at CyTOF2 detector. Figure 1 a of the supplemental information shows, however, that 1500 nanoparticles per cell generate mean raw signal (Au197Di, which is dual counts at the detector) of close to 1500, e.g. one count per one nanoparticle. There is a factor of ~ 10 discrepancy here. Is the horizontal axis showing dual counts, or the number of nanoparticles? Could you please also propose explanation (or suggest that there is no clarity yet, since this is a communication) why the distributions for 5, 8, 23 and 66 nanoparticles per cell are bimodal?

We made our calculation based on the transmission factor for ^{193}Ir , as the closest atomic weight element to Au that can be used for calibration of the instrument. The transmission factor accounts for the percent ionization, and our calibration for the detection limit experiment gave us:

$$^{193}\text{Ir transmission factor} = (\text{dual counts in tuning solution}) / (\text{atoms introduced}) = (791225) / (9.78 \times 10^8) = 0.000809$$

Note that atoms introduced = (Ir tuning solution concentration)*(integration time)*(flow rate)*(natural abundance of ^{193}Ir)*(Avogadro's number)/(^{193}Ir isotope mass) = 9.78×10^8

We assumed for our detection limit calculations a NP size of 3 nm diam., and that the ionization of Ir and Au are approximately equivalent (Degree of ionization is proportional to the ionization energy of an element, and Ir and Au have ionization energies of 8.9760 eV and 9.2255 eV, respectively). Therefore:

$$10 \text{ NP} = 8340 \text{ Au atoms} = (\text{dual counts})/(\text{transmission factor})$$
$$\text{dual count of } ^{197}\text{Au} = (8340 \text{ Au atoms})(0.000809) = 6.75$$

For the supplemental Fig. 1a condition mentioned, the mean raw signal was 1022 dual counts, giving by the same calculation:
 $(1022 \text{ dual counts})/(0.000809) = 1.263 \times 10^6 \text{ Au atoms} = 1514 \text{ NPs}$ which is self-consistent.

Although the ionization energies of Ir and Au are very close, using Ir transmission to convert gold dual counts to gold ions constitutes a potential source of error. Therefore, we used ICP-AES to reassure that the mass cytometry readout of the relative quantity of Au NPs per cell was valid- these comparisons suggest this Ir calibration is reasonable. We have edited the manuscript on p.3 and clarified the calculations in the Methods on p.8 to make these points more clear.

As for the bimodal populations observed at low total particle concentration, on examining the raw data/gating more closely, we discovered this was an artifact of the broad gate we used in the initial identification of cells by DNA staining- this has been corrected and is discussed in detail below under Reviewer 2 point 2.

9) The abstract would benefit from inclusion of the vaccine delivery experiment result (tumor outgrowth control).

We have updated the abstract to comment on these results.

Impressive work, keep up!

We thank the referee for these positive comments.

Reviewer #2 (Remarks to the Author):

The paper by Yang et al studied 3 nm gold nanoparticles and their interaction with cells. Initially, they used BODIPY-labeled gold NP that were taken up nonspecifically by cells, and demonstrated that the amount of gold could be accurately measured by CyTOF. Similarly, the signal was in proportion to the concentration of NP used; the use of ICP-AES helped confirm proportionality.

Later, the authors extended the work to show that the NPs were taken up to different extents by cells present in lymph node tissues. They demonstrated that myeloid dendritic cells accumulated the most. Injecting antigen-conjugated NPs into mice allowed much greater survival against tumor challenge than free antigen or linker-antigen. This could be of great use as a general method of vaccination/antigen-presentation in the future: it would have the advantage of being a well-characterized particle to which virtually any antigen could be conjugated.

Nanoparticle conjugations and their introduction into animals or cells isn't new, but I do feel that the authors did a good job of characterizing their construct, and demonstrating effectiveness in the tumor challenge model. Their particles are also "visible" by CyTOF, which could help multiparameter characterization of other, future constructs.

Major issues:

1. There are no calculations stating the approximate number of Au ions per particle. This has great importance for the CyTOF: the detector can be saturated, which is a likely reason for the CyTOF line deviating from the ICP-AES line in Fig 2c, and the lower-than-expected signal in Fig 2a for 100 ug/mL.

The number of atoms per particles was calculated as follows: Considering the NPs as monodisperse spheres, the number of face-centered cubic (FCC) cubes per 3 nm spheres can be calculated (gold lattice constant 4.080 Å). Each FCC cube contains $8 \cdot (1/8) + 6 \cdot (1/2) = 4$ atoms, thus the total number of Au atoms per 3 nm sphere can be determined. (For example, a 3 nm AuNP contains 834 atoms). CyTOF2 and Helios instruments have a dynamic range of 4.5 orders of magnitude, but indeed, the lower-than-expected signal in Fig 2c is caused by detector saturation. We have updated the Methods on p.8 to comment on these calculations explicitly.

2. I am not convinced that their technique can actually accurately, reproducibly detect down to 10 particles/cell as stated on Line 87. Since the authors used a cell line (rather than a mixture of cells like PBMCs) for experiments like Fig 2 and Suppl Fig 1, I would expect a consistent, unimodal cell labeling.

Looking at Fig 2a, the 0.01 ug/mL curve in dark green is not unimodal; there is a population with zero Au signal. The 0.1 ug/mL - 10 ug/mL curves are unimodal; the 100 ug/mL probably is too, but appears to be saturating the detector. Looking at Suppl Fig 1, the lowest concentrations (0.00125-0.01 ug/mL; 5.33-65.63 NP/cell) are completely bimodal.

I will agree that they can properly, reproducibly detect down to 0.1 ug/mL, but that corresponds to 1515 NP/cell, not 10 NP/cell. This is still substantially better (15-20x?) than the fluorescence data in Fig 2a, so I don't see this as any kind of failure.

We agree the presence of multiple peaks in the Au NP uptake histograms was confusing. Prompted by the reviewer's comment we examined the raw data more closely and discovered that these non-monomodal profiles were an artifact of the broad gate used to select cells in the mass cytometer data analysis. As shown in the **Figure R1** for the reviewers below, the first step in gating the CyTOF data is to select cell-based events by gating on events double-positive for bright staining with a pair of DNA labels (proprietary DNA intercalators conjugated with two different isotopes of Ir), which is intended to avoid analysis of non-cellular debris. We discovered that the wide DNA⁺ gate used in the original figure included multiple peaks, that we suspect may reflect cellular debris rather than the main peak of intact cells. A more conservative gate focused on the main DNA⁺ population gives a distribution that looks as expected, with a proportion of nanoparticle-negative cells and a monomodal population of cells that have taken up particles. We suspect that the wide gate we originally used is capturing exosomes or other cellular debris that take up the DNA stains but are not actual cells.

Figure R1. Revised CyTOF gating for cell identification in detection limit experiments. (a) RAWblue cells were treated with 0.01 ug/mL AuNPs for 6h at 37°C and analyzed on a Helios mass cytometer. Events double positive for the ¹⁹¹Ir and ¹⁹³Ir DNA stains were first identified. (b) Initial broad gating on DNA stain used in our original analysis (yellow) vs. restricted gating on the major event population (red). (c, d) ¹⁹⁷Au dual counts histograms obtained by the initial wide gating (c) and revised main population gating (d), the latter excluding a low DNA-stain intensity event population that may be cellular debris.

We re-analyzed the detection limit data using this revised gating strategy and the data is shown in the revised Fig. 1C and Supplementary Fig. 1. Our original analysis also made calculations of mean particles/cell based on gating on only NP⁺ cells (i.e. we plotted the mean particle concentration for cells that had >0 NPs); in the revised analysis we corrected this calculation by determining the mean particles/cell including the NP-negative population, which is significant at the lowest particle concentrations. The conclusions are not greatly affected: At high NP concentrations the mean particles/cell is only modestly altered. At low concentrations, this corrected analysis reveals that the lowest two tested particle concentrations give statistically not-significant dual counts, and the detection limit occurs at a dose of 5 ng particles/mL, corresponding to 8±2 particles/cell on average. (Note that at this particle concentration, the histogram has its peak for NP⁺ cells at ~10 dual counts ¹⁹⁷Au per cell, corresponding to ~15 gold particles). We feel this revised analysis is correct and captures the results accurately. We

have revised the text on p. 3 and Methods on p. 8 to clearly convey the gating approach and analysis for the detection limit determination, keeping the claim that the limit is ~10 particles per cell based on these findings.

As an aside, Fluidigm claims the Helios instrument used here has a cell identification lower detection limit of 350 antibodies per cell, and each antibody is tagged with ~40-100 metal isotopes, which means 14,000-35,000 atoms per cell. This is in reasonable accord with our finding that NP⁺ cells at the detection limit containing ~15 particles = 12,510 Au atoms.

Minor issues:

1. On page 1, lines 37-39, the authors talk about how "most nanoparticles lack intrinsic fluorescence". However, Qdots (particularly Cd-based ones) have been used in fluorescence flow and microscopy for decades. Also, the Bendall et al reference 16 used a CD3-Qdot in their CyTOF experiment.

Similarly, recent work detailing how different-sized gold nanoparticles can have intrinsic color different than that of bulk gold metal due to surface plasmon resonance and such seems relevant. This seems especially important, since several are commercially-available from Sigma, including some in the 5nm range that would be relevant to the 3nm particles detailed in the article.

This statement was not meant to imply that there are no intrinsically-fluorescent nanoparticles- hence "most" (there are dozens of other common nanomaterials that have no intrinsic fluorescence). However, we have edited the statement to: "...for nanoparticles that lack intrinsic fluorescence, a fluorescent tag must be attached."

2. In the caption for Fig 1c, the authors state "Size distributions and zeta potentials.... determined from TEM and dynamic light scattering, respectively". DLS doesn't give zeta potential. The Zetasizer NanoZS used by the authors is capable of doing both DLS and zeta potential, but they aren't the same: according to the manufacturer's website, the DLS is done by non-invasive back-scatter, while laser doppler micro-electrophoresis with PALS is used for zeta potential.

Aggregation state mentioned in line 186 should be measurable, at least in a general sense, by DLS looking at size distributions.

We corrected the caption to indicate the Zetasizer instrument, rather than DLS per se, was used for zeta potential measurement. We agree DLS can be useful for detecting particle aggregation; The comment on line 186 refers to the inability of

mass cytometry itself to track this property.

3. While Au197 is the only naturally-occurring, stable gold isotope, I would still recommend stating the mass measured in places such as Fig 2a right plot.

This is a good suggestion. We have added ^{197}Au to the axe labels.

Reviewer #3 (Remarks to the Author):

This manuscript demonstrates the use of CyTOF technology and metal-tagged antibodies to determine the cellular fate of nanoparticles *in vivo*. The manuscript is not the first to report quantification of nanomaterials in single cells using ICP-MS based techniques (see e.g. Zheng, Talanta, 2013, 116:782), nor is it the first to co-detect nanoparticles and metal-tagged antibodies (see e.g. Managh, J. Immunol., 2014, 193:2600). However, the highly multiplexed nature of the work presented here (13 parameters) represents an advance over the existing literature, particularly in relation to the ability to quantify and correlate nanoparticle uptake with cellular phenotype. As is demonstrated for vaccine delivery, the method has the potential to influence the development of inorganic nanomedicines and thus will be of interest to workers in this field.

Overall the manuscript is presented in a clear and concise format, the data is well presented and the main conclusions are adequately justified by the data.

Specific points to be addressed:

1) Abstract: The authors claim that the technique is "applicable to diverse types of inorganic nanomaterials", yet the discussion within the paper does not provide sufficient information to support this statement. For instance, could the technique be applied to elements that have a higher background *in vivo* than Au? The authors should also comment on the mass range of the instrument (75-209 amu) in relation to this.

We have edited the Discussion on p. 5-6 to discuss this issue briefly:

“This method should be applicable to the sensitive detection of many other inorganic nanomaterials ranging from 75-209 amu, including elements already used in biomedical nanoparticles such as platinum, bismuth, gadolinium, lanthanides, and palladium. Exceptions will likely be metals that have high endogenous concentrations *in vivo*, such as molybdenum and selenium, though the mass range of current mass cytometry instruments was explicitly designed to exclude elements prevalent *in vivo* (generally < 75 amu), which would otherwise saturate the TOF detector.”

2) Introduction: The introduction should make reference to previous work in the field of single particle and single cell ICP-MS, and its relationship to detecting nanomaterials within cells using the CyTOF.

We added a paragraph to discuss single cell ICP-MS and single particle ICP-MS in the introduction on p. 2 along with supporting references.

3) Page 2 Line 54: The term "label-free method" (used throughout the manuscript) could potentially be misleading, given that antibody based labels are subsequently used to identify the cellular destination of the nanoparticles. Consider rephrasing.

We have altered the text where appropriate to indicate the method is label-free for detection of the nanoparticles only- this is an important advantage. Addition of labeled antibodies for *ex vivo* analysis is not accompanied by the same issues as introduction of a label to the particles that must survive and not disrupt the behavior of the particles *in vivo*.

4) Page 2 Line 81: Was "the upper detection limit of $\sim 1 \times 10^6$ particles/cell" imposed by detector saturation? This warrants further discussion. As the upper limit is a function of the total metal content per cell, what might this mean for the detection of larger i.d. nanoparticles than the relatively small (~ 3 nm) ones studied here? Note that the tail of the 100 $\mu\text{g/ml}$ peak in Figure 2a appears to have been cut off. Were these cells not detected due to detector limitations?

This is an excellent point and we have revised the text to discuss this issue. The referee is correct, at the highest concentrations of gold treatment *in vitro* the deviation of the mass cytometry data from ICP-AES measurements reflects detector saturation in the CyTOF. For the instruments we used in this study, saturation was reached at approximately 1.32×10^9 gold atoms per cell, corresponding to 1.5×10^6 nanoparticles 3 nm in diam., 42,906 particles 10 nm in diam., 12,713 particles 15 nm in diam., 343 particles 50 nm in diam., or 43 particles 100 nm in diameter. Thus, the dynamic range will be highest for the smallest particle sizes. These limitations still make the method quite valuable since gold particles in biomedical applications are most commonly ≤ 50 nm in diam. We have commented on this issue in the revised discussion on p. 6.

5) Page 3 Line 102: Gold is known as a highly retentive element in the CyTOF system. Can the authors be sure that detection of gold in "virtually all of the cells" was not due to memory effects?

We checked that there was no memory effect by analyzing gold-negative control

cells before and after all gold-containing sample runs, and found that Au signal of control cells was as low as the initial background value (nearly zero). We have commented on this finding in the revised text on p.3.

6) Page 5 Line 181: A comment on the total analysis time (and time taken for cellular staining), in comparison to the techniques mentioned in the introduction, would be helpful.

We have added a comment about analysis time to the discussion on p.5.

7) Figure 1: The label (d) is in the figure, but not referred to in the caption.

We corrected this error.

8) Figure 2c: The error bars mentioned in the legend are not visible on this figure. If the s.e.m. is too small to be observed, another measure e.g. s.d. might be more easy to visualise.

We have modified Figure2c to show s.d. instead of s.e.m. We have also updated the figure with corrected ICP-AES readout values (previous plots showed raw data without accounting for a dilution factor).

9) Page 7 Line 260: Despite the section title, it is not clear how the number of Au atoms per cell is converted to the number of gold nanoparticles per cell. Either the formula used should be included or an appropriate reference provided. An estimate of the uncertainty in the conversion should also be included somewhere in the manuscript.

We have added a detailed description of the calculation to the revised methods on p.7-8, as discussed in the response to reviewer 2.

Reviewer #1 (Remarks to the Author):

This referee is satisfied with corrections and revisions done in response to all comments and suggestions.

There are two additional suggestions:

1) The minor correction is needed for the added part of the Discussion, where authors mistakenly call selenium a metal: "Exceptions will likely be metals that have high endogenous concentrations in vivo, such as molybdenum and selenium,..."

Selenium is a non-metal, please correct.

2) One of the referees suggested a reference to the detection of nanomaterials in single cells by ICP-MS as Zheng et al, *Talanta*, 2013. This referee suggests that in this context, an article published in 2007 should be cited. In Tanner et al., *Spectrochimica Acta B*, 62 (2007), 188-195, exploration of single cell assay via ICP-MS was presented, with detection of nanogold labels of secondary anti-CD34 antibody in single myeloblast cells.

Thank you for highlighting the changes in the revised manuscript.

Congratulations on a very good work!

Reviewer #2 (Remarks to the Author):

As Reviewer 2, my comments from the original manuscript have been sufficiently addressed by the authors.

Reviewer #3 (Remarks to the Author):

I am satisfied that this updated manuscript fully addresses the concerns raised in the reviewer report and am pleased to recommend acceptance of the manuscript.

I congratulate the authors on an interesting piece of work.

Nature Communications 2016 Yang et al.

Reviewer #1 (Remarks to the Author):

This referee is satisfied with corrections and revisions done in response to all comments and suggestions.

There are two additional suggestions:

1) The minor correction is needed for the added part of the Discussion, where authors mistakenly call selenium a metal: "Exceptions will likely be metals that have high endogenous concentrations in vivo, such as molybdenum and selenium,.."

Selenium is a non-metal, please correct.

This error is now corrected. We have replaced "metals" with "elements" in the main text on p. 5.

2) One of the referees suggested a reference to the detection of nanomaterials in single cells by ICP-MS as Zheng et al, *Talanta*, 2013. This referee suggests that in this context, an article published in 2007 should be cited. In Tanner et al., *Spectrochimica Acta B*, 62 (2007), 188-195, exploration of single cell assay via ICP-MS was presented, with detection of nanogold labels of secondary anti-CD34 antibody in single myeloblast cells.

We have referenced *Tanner et al., Spectrochimica Acta B*, 62 (2007), 188-195 in addition to Zheng et al, *Talanta*, 2013 in the main text on p. 2.

Thank you for highlighting the changes in the revised manuscript.

Congratulations on a very good work!

We thank the referee for these positive comments.

Reviewer #2 (Remarks to the Author):

As Reviewer 2, my comments from the original manuscript have been sufficiently addressed by the authors.

We thank the referee for these positive comments.

Reviewer #3 (Remarks to the Author):

I am satisfied that this updated manuscript fully addresses the concerns raised in the reviewer report and am pleased to recommend acceptance of the manuscript.

I congratulate the authors on an interesting piece of work.

We thank the referee for these positive comments.

Reviewer #1 (Remarks to the Author):

This reviewer is fully satisfied with the revised manuscript, recommend publication.

Congratulations on good work!